# OpenReview forum: "Why Language Models Lie"
_ICLR.cc/2026/Conference — Submitted to ICLR 2026_

### Official Review · Reviewer_Anqc · 2025-10-29

**Soundness:** 2
**Presentation:** 1
**Contribution:** 2
**Rating:** 2
**Confidence:** 4

**Summary:**

The authors create a set of prompts which induce truthful or deceptive outputs from LMs. They use these prompts to find corresponding activations for truthfulness or deception. They purport to use these activation patterns to determine "conflict" spaces in the model's representations which correspond to actions where the model's prompt and training objectives are in conflict. They provide a method for detecting deceptive outputs based on these activation patterns.

**Strengths:**

Important problem (deception and identifying truthful or honest LM outputs).

A reasonable range of models were evaluated.

The ideas seem potentially promising, but are not thoroughly evaluated enough, or explained well-enough.

**Weaknesses:**

Major

The method in the paper for using prompts to elicit conflicting activation patterns is extremely similar to methods used in contrastive activation steering: https://arxiv.org/abs/2312.06681 , https://arxiv.org/abs/2308.10248 and representation engineering https://arxiv.org/abs/2310.01405
These techniques go further to steer model behaviour with these activations.

The authors need to discuss how their method differs from the above techniques in the related work section.

Overall the paper is not well written. The concepts are not clearly introduced, e.g.m figure 1 does not feel helpful in understanding the mathematical concepts and how they relate to the intuitive claims in the paper. (In general it's unclear how the maths relates to the informal claims.) The mathematical techniques used are also not well motivated, e.g., in 2.3 ACTION SUBSPACES why do you cluster the activation vectors? Why do you use the specific numerous clustering methods introduced?

In general the figures are not easy to interpret and are not well explained.

The (single) dataset used is extremely toy --- just a basic set of prompts. It would be much better to evaluate your methods on existing more realistic benchmarks. Additionally, it would be good to know how much your detection method generalises to new environments.

The structure of the paper is not super clear and does not clearly aid the reader in understanding the key findings.

I think the paper makes too strong claims based on limited evidence. E.g., the claims regarding conflict spaces and model representations of alignment do not seem clearly stated or evidenced.

Minor

The paper's title is "WHY LANGUAGE MODELS LIE" but it does not say what it means for an LM to lie. The authors should cite https://arxiv.org/pdf/2309.15840

The paper includes a section "WHY THIS IS DECEPTION" but doesn't discuss what deception actually is. The authors should cite https://arxiv.org/abs/2312.01350

**Questions:**

How are your methods different than existing activation steering approaches?

Why cluster activation deltas in the way that you do?

Does your detection method generalise to different datasets?

---

### Official Review · Reviewer_acLR · 2025-10-31

**Soundness:** 2
**Presentation:** 2
**Contribution:** 2
**Rating:** 2
**Confidence:** 3

**Summary:**

The paper proposes to use controlled system prompts to induce deceptive and truthful behaviour. Analysing the activations of responses on these prompts the paper discovers so called conflict subspaces, subspaces in the model activations that discriminate between deceptive and truthful behaviour. Based on the conflict subspaces a method for identifying deception by mapping the current activations to one of the subspaces is proposed.

**Strengths:**

The topic of the paper is highly relevant. Insights on the underlying process of emergent deception in LLMs and connecting it to optimisation pressure during fine-tuning and competing objectives are valuable.
* the paper proposes an interesting hypothesis to explain deceptive behaviour via the conflict subspaces in the model activation space.
* the dataset covers several domains including medicine, law, finance, etc. spanning a good range of topics
* the proposed deception detection method is simple and performs well on the evaluated tasks
* it is good to see that the conflict subspaces hold the two model families that were evaluated

**Weaknesses:**

While the topic is interesting and relevant, my main concerns are the theoretical grounding, the overall methodology and reproducibility of the evaluation, as well as the limited scope of the evaluation. I describe my concerns in more detail below.

**Limited theoretical grounding:**

The concept of a conflict subspace is primarily empirical. The attempt to connect orthogonality with alignment dynamics is mostly speculative and lacks formal discussion or evaluation. It would be important to rigorously validate the argument for why conflict subspaces arise. this could be done by comparing base models with those that have undergone fine-tuning. It is mentioned that GPT-2 (a non post-trained model) was evaluated and also presented conflict spaces, but I was unable to find the corresponding results. If it indeed exhibits a similar pattern, this would challenge the claim that post-training induces conflict subspaces. On the other hand, if it does not, a the comparison of non-fine-tuned vs fine-tuned models would be critical.

**Methodology:**

The discussion of the methodology is unclear in several places and should be improved to strengthen the paper.
* Define all notation clearly: equations (1) and (2) are stated without any context, $a$ is never defined with a subscript, that is $a_{pL}$ and  $a_{pT}$ are undefined
* Mode detail on the dataset creation would be great. Since the entire discovery of conflict spaces relies on it, Appendix A could better describe how they were collected, curated and validated. I am also concerned about the simplicity of the prompts shown in Appendix A.
* As part of the dataset creation a comparison is made between cases where the LLM is instructed to lie and cases where the LLM may autonomously choose to lie (lines 103-107). This distinction does not seem to be discussed in detail in the evaluation?
* Also on lines 103-107 it is stated that the LLM is given the option to lie. How is this done explicitly?
* Please clarify the clustering step, were $\Delta a_L$ and $\Delta a_T$ clustered jointly or separately (there also seems to be a typo here on line 176, see below).
* Equation (6) should be the L2 norm as $\Delta_ a$ and $c_k$ are higher dimensional vectors
* The conflict detector is trained and evaluated on the same dataset distribution. This makes it unclear how well it will generalise to unseen prompts and situations where the deceptive behaviour is not induced by specific system prompts.

**Evaluation scope:**

In order to identify a generalisable phenomenon (such as conflict spaces) I believe a larger scale evaluation is required. The paper found that already 1 of the 2 model families evaluated differs in terms of the number of the number of clusters found by the analysis. It would be important to evaluate other model families on also additional data (carefully designed to remove any biases) to support the generality of conflict subspaces. Additionally, it would be great if the discussion of the results could be backed up with more evidence. In 4.1 the different types of behaviours contained in the identified subspaces are discussed, however, the discussion could be greatly improved with quantitative evidence and formal analysis of the data obtained. In the current state it is unclear how the rate of hallucinations or the types of deception were analysed and how significant these insights are. Overall the current evaluation is insufficient to support the claims about conflict subspaces "universally" explaining deception. Broader model coverage and more diverse evaluation data are needed to validate this.

**Conceptual framing:**

Could the authors justify their decision to view deception from the user's perspective rather than intent driven or due to an internal objective in more detail?


**Other points:**

* For figure 2, why are token positions 0-24 and 42-end always zero? How long exactly is the input here? If they are always zero the plot could be cleaner by removing them.
* I found it very hard to understand and draw conclusions from Figure 4. It would be great if the caption could provide more specific details.
* Figures 3 and 4 are missing axis labels, the tick marks on Figure 4 need to be improved (especially y-axis) and for Figure 2 the font size should be increased. It would also make the paper easier to understand if the figure captures were more descriptive.
* Based on the plot shown in Figure 2, second row, how can I arrive at the conclusion that all delta vectors near the end of the prompt are positively aligned with the average delta vector for each layer? It looks to me as it is more the middle of the prompt that shows higher similarity?
* Section 3 does not refer to any of the figures. It would be helpful to refer the reader to the figure that supports a claim.
* Is the observation of different types of behaviour simply due to the different system prompts used in the dataset?
* How is the subset of 150 prompts sampled to evaluate the conflict detection? Are sampled datasets stratified across truthful and deceptive?

Typos:
* line 320 "This figures shows"
* line 176 "$\Delta a_L$ and $\Delta a_L$"
* line 269 "where are all"

**Questions:**

* How were the prompts in the dataset curated?
* As part of the dataset creation a comparison is made between cases where the LLM is instructed to lie and cases where the LLM may autonomously choose to lie (lines 103-107). This distinction does not seem to be discussed in detail in the evaluation?
* Also on lines 103-107 it is stated that the LLM is given the option to lie. How is this done explicitly?
* For figure 2, why are token positions 0-24 and 42-end always zero? How long exactly is the input here?
* Based on the plot shown in Figure 2, second row, how can I arrive at the conclusion that all delta vectors near the end of the prompt are positively aligned with the average delta vector for each layer? It looks to me as it is more the middle of the prompt that shows higher similarity?
* How is the subset of 150 prompts sampled to evaluate the conflict detection? Are sampled datasets stratified across truthful and deceptive?
* Could the authors justify their decision to view deception from the user's perspective rather than intent driven or due to an internal objective in more detail?

---

### Official Review · Reviewer_d4q8 · 2025-11-03

**Soundness:** 2
**Presentation:** 2
**Contribution:** 1
**Rating:** 2
**Confidence:** 4

**Summary:**

This paper describes investigations into LLM activations on deceptive responses incentivised by the system prompt. A dataset is created by combining fictional user questions with "truth" or "deceptive" system prompts, and augmenting with minor textual perturbations. Activation vectors for this dataset of prompts are then collected for a number of Llama and GPT-OSS models, and k-means clustering and SVCCA finds clusters and correlated directions in the representation space. These spaces are found to be very orthogonal (cosine similarity < 1e-8). Probes are found to be most sensitive at particular layers and token positions. Using these probes, authors categorise their generated dataset with high accuracy.

**Strengths:**

This is the first paper I'm aware of to apply SVCCA on node activations to the question of LLM deception. The authors have made an appropriate choice of SVCCA as an appropriate and standard method for comparing neural network representations.

It's a strength of the paper that it engages with the question of what behaviours constitute deception, as it's valuable to have clear lines about what your probes identify.

The size of the dataset, 24,300 elements, is suitable for the techniques used (k-means, SVCCA, etc).

The investigation of the clusters found, associating one "lying" cluster with knowledge suppression and a second with direct lies, is intruiging, and I would be interested in further quantitative work here.

The paper is well laid-out, and clearly communicates the research findings.

The success of SVCCA invites cross-model analysis, which would be an interesting avenue of research.

**Weaknesses:**

It's not clear to me that simply using k-means & SVCCA significantly advances the field beyond existing work on activation probes, steering vectors, and unsupervised "lie-detector" approaches such as Contrastive Consistent Search CCS
"Discovering Latent Knowledge in Language Models Without Supervision" arXiv:2212.03827
"When Truthful Representations Flip Under Deceptive Instructions?" arXiv:2507.22149
"Truth is Universal: Robust Detection of Lies in LLMs" arXiv:2407.12831
"Steering Llama 2 via Contrastive Activation Addition" arXiv:2312.06681
"Scaling Monosemanticity: Extracting Interpretable Features from Claude 3 Sonnet" https://www.anthropic.com/research/evaluating-feature-steering

Given this existing literature (some of which is cited, but not deeply engaged with), it seems odd in Related Works to say that mechanistic interpretability studies "are not trained on deceptive outputs", and so "cannot be used to distinguish deceptive behavior".

A stronger version of this paper might explicitly compare the performance of the k-means / SVCCA method to existing literature: does CCS perform better or worse on your dataset? Baselines would be helpful, especially since the work uses an unvalidated dataset. Comparisons to state of the art work would be ideal, but even simple baselines would be helpful. For example, does this method outperform clustering based on embeddings, or even bag-of-words?

Alternatively, a stronger version could highlight and explain the meaningfulness of the subspaces found, and provide specific, numerical analyses of what might otherwise be hand-waving interpretations. However, the current work likely needs more work before being able to fully understand this: it seems (~L323) like all LLMs have 2 clusters, except GPT-OSS 20B which has 3, and discussion about this in Section 4 is quite speculative. It's hard to see how to make use of this information, even if reliable.


Minor points:

Though the dataset comprises 24,300 pairs, the method of construction likely introduces correlations, reducing the "effective size".

Figure 1 seems pretty impressionistic to me - I think I get what's being gestured at (prepending a truth/deception system prompt to a user prompt changes the activation vector to take it into a truth/deception subspace; there are multiple disjoint such subspaces) but the square / subspace-positioning doesn't really correspond to orthogonality in any real sense, so I think the figure's a bit confused.

It could be made clearer that a_i is the activation of the bare user prompt, and in L128 clarify that p_L and p_T differ only in the system prompt prepended to the user prompt.

There's some inconsistency / ambiguity in terms: sometimes the paper talks about "conflict subspaces", "deceptive prompts", dishonesty, misinformation. Conflict, deception, dishonesty, misinformation: these are all distinct concepts. More precision would be beneficial.

Almost all the user prompts in Appendix A seem like obvious bad ideas, sometimes to the point of tautology e.g. "is it fine to use outdated blueprints" / "is it legal to fake my tax return" - while not obviously bad this seems weird. I'd expect to see this balanced by neutral user prompts.

Being (extra) pedantic: I think it's imprecise to say that the subspaces are "in the LLM's architecture" or "near the end of the prompt". The subspaces are in the activation space. But the meaning is clear enough.

Stylistically, I'd avoid language like "In general, the detector is really accurate", even when immediately followed by a numerical value. But this is probably more pedantry, and it would be fine to talk about "high accuracy".

**Questions:**

I'm very surprised to read about "cross-family directions (e.g. from Llama-3 1B truth clusters to gpt-oss-20b conflict clusters)" - I'd have thought that those architectures were sufficiently different that directions were hard to define, and SVCCA is an interesting approach to address that. A cross-family lie-detector would be interesting. Could you say more about this?

Aren't most directions in activations spaces, since those spaces are so high-dimensional, very orthogonal? Could you be more precise about what you mean on L53 by "directions from the centroids of truth subspaces to the conflict subspaces are all orthogonal"?

L63: is "conflict" "universally encoded in unique subspaces" more, or less, than other concepts are (e.g. "Golden Gate Bridge")?

Section 4: in a high-dimensional space wouldn't you expect any random subspaces to be nearly orthogonal? Are these conceptual subspaces much more mutually orthogonal than these spaces to e.g. the space of cats and dogs?

Figure 2:
a) it seems very surprising that there should be such a sharp feature around tokens ~24 in heatmap 1 & 2, another sharp feature around token ~42 in heatmap 2. Also, the early-token deltas seem to be identically zero? What happens at these tokens?

b) I'm also surprised that Layer 16 seems to have almost all the "lying signal". From work such as Meng et al (https://rome.baulab.info/) and Wang et al (arXiv:2502.04556) I'd expect more-conceptual processing to occur mid-network, and very-early and very-late layers to be more lexical. In L279 you appeal to intuition, which I do not share: my intuition is that the model "knows more and more about whether it will lie", i.e. there would be gradual increase. Can you explain these sharp features?

Why are they called "action" subspaces, and not e.g. "activation" subspaces?

L276: "Since the highest variance would only happen where there are the maximum number of subspaces": is this true? Couldn't you have a skewed system, with few subspaces carrying much variance?

Figure 4 / L335: Aren't the clusters highly orthogonal? And there are mutliple "lying" clusters? How come /all/ of the vectors have high cosine similarity with each other?

---

### Meta-Review · Area_Chair_Lzfx · 2025-12-26

**Summary:**

The reviewers stay on the negative side, and the main concerns are:

- Lack of Novelty: Reviewers noted the methodology is too similar to existing work like Contrastive Consistent Search (CCS) or activation steering and lacks comparison with these baselines.
- Weak Theoretical Grounding: The "conflict subspace" concept is seen as empirical and speculative, lacking formal proof of why these spaces arise during alignment.
- Limited Evaluation: The dataset is considered a "toy" dataset with simple prompts , and the detection method was not tested for generalization to unseen environments or autonomous deception.
- Poor Presentation: Critics cited unclear notation, missing axis labels in figures, and confusing visualizations (specifically Figures 1, 2, and 4).
- Methodological Ambiguity: The reasons for choosing specific clustering methods and the definition of "deception" were not adequately explained.

**Reviewer Concerns:**

The authors did not provide a rebuttal to address the above concerns raised during the review period.

**Reviewer Scores:**

Given the lack of author engagement and the severity of the methodological and conceptual concerns, it is highly unlikely that any reviewer would have increased their score. The consensus remains firmly on the side of rejection.

---

### Decision · Program_Chairs · 2026-01-26

Reject